# New-Generation Glucokinase Activators: Potential Game-Changers in Type 2 Diabetes Treatment

**DOI:** 10.3390/ijms25010571

**Published:** 2024-01-01

**Authors:** Dania Haddad, Vanessa Sybil Dsouza, Fahd Al-Mulla, Ashraf Al Madhoun

**Affiliations:** 1Genetics and Bioinformatics Department, Dasman Diabetes Institute, Dasman 15462, Kuwait; dania.haddad@dasmaninstitute.org (D.H.); vanessa.dsouza96@gmail.com (V.S.D.); fahd.almulla@dasmaninstitute.org (F.A.-M.); 2Animal and Imaging Core Facilities, Dasman Diabetes Institute, Dasman 15462, Kuwait

**Keywords:** dorzagliatin, TTP399, glucokinase activator, type 2 diabetes, clinical trials, safety, efficacy

## Abstract

Achieving glycemic control and sustaining functional pancreatic β-cell activity remains an unmet medical need in the treatment of type 2 diabetes mellitus (T2DM). Glucokinase activators (GKAs) constitute a class of anti-diabetic drugs designed to regulate blood sugar levels and enhance β-cell function in patients with diabetes. A significant progression in GKA development is underway to address the limitations of earlier generations. Dorzagliatin, a dual-acting GKA, targets both the liver and pancreas and has successfully completed two phase III trials, demonstrating favorable results in diabetes treatment. The hepato-selective GKA, TTP399, emerges as a strong contender, displaying clinically noteworthy outcomes with minimal adverse effects. This paper seeks to review the current literature, delve into the mechanisms of action of these new-generation GKAs, and assess their efficacy and safety in treating T2DM based on published preclinical studies and recent clinical trials.

## 1. Introduction

Type 2 diabetes mellitus (T2DM) is a chronic metabolic disease characterized by impaired glucose homeostasis. The pathophysiological mechanisms involved include insulin resistance (IR) and a reduction in glucose-stimulated insulin secretion (GSIS), accompanied by elevated fasting plasma glucose (FPG) levels and increased hepatic glucose production (HGP) [1,2]. Chronic hyperglycemia is associated with a higher risk of complications, including cardiovascular disease, retinopathy, neuropathy, and nephropathy [3,4]. To address the multiple pathogenetic disorders observed in diabetes, a combination of several anti-diabetic agents is often necessary to achieve optimal glycemic control. Current oral therapies encompass insulin and its analogues, sulfonylureas, biguanides, glinides, thiazolidinediones, alpha-glucosidase inhibitors, glucagon-like peptide 1 receptor agonists, dipeptidyl peptidase 4 inhibitors, and sodium-glucose cotransporter 2 inhibitors [5,6,7,8,9,10,11,12]. However, these therapies have limitations, such as insufficient efficacy or the potential for adverse events (AEs), including hypoglycemia, weight gain, gastrointestinal discomfort, urinary tract infection, or heart failure [13,14,15,16]. Furthermore, these therapies do not adequately address the primary concern of progressive β-cell failure [17,18,19]. Nonetheless, ongoing efforts to develop novel treatment modalities have led to the introduction of new medications. The glucokinase activators (GKAs), dorzagliatin and TTP399, represent a promising class of anti-diabetic drugs, with the potential to maintain glucose homeostasis.

## 2. An Overview of GK and GKAs

Glucokinase (GK) plays a crucial role in glucose metabolism, catalyzing the rate-limiting phosphorylation of glucose into glucose-6-phosphate (G6P) [20]. This step is essential for ATP generation, energy production, and cellular homeostasis. In the pancreas, GK serves as a glucose sensor in β-cells, directly regulating insulin secretion [21]. As blood glucose levels rise, the increased kinetic activity of GK mediates G6P production, subsequently stimulating insulin release and maintaining glucose homeostasis [22]. In pancreatic α-cells, GK regulates glucagon secretion [23]. Unlike β-cells, GK activity in α-cells is not directly responsive to glucose levels, but rather regulates amino acids and fatty acids [24]. GK is also expressed in various other tissues, including the liver, enteroendocrine cells, adrenal glands, neurons, and anterior pituitary cells [25,26,27]. In hepatocytes, GK is the key enzyme for glyconeogenesis, the process of converting non-carbohydrate precursors into glucose [28]. Thus, GK is a versatile enzyme that plays a vital role in glucose metabolism and homeostasis in various tissues [26].

The association of T2DM with pancreatic GK activity is not well-established [29]. An increase in GK activity has been shown in diabetic patients and obese individuals with diabetes [30,31]; however, it is also observed to be reduced in patient with newly diagnosed T2DM [32]. The regulation of GK is multifaceted and occurs at both transcriptional and post-transcriptional levels [33]. During fasting, hepatic GK is sequestered into the nucleus in an inactive form, forming a complex with the inhibitory protein glucokinase regulatory protein (GKRP) [34]. In response to glucose intake during the postprandial period, the GK/GKRP complex dissociates, allowing GK to translocate to the cytoplasm and become active [34]. In T2DM, alterations in glucose are postulated to reduce GK gene (*GCK*) expression, impair compartmental shoveling by GKRP, and/or dysregulate other metabolic and hormonal conditions that regulate GK [35,36], as best reviewed in [33].

In humans, genetic polymorphisms in the *GCK* gene are associated with a mild hereditary form of diabetes known as maturity-onset diabetes of the young (MODY) 2 or GCK-MODY. GCK-MODY manifests as mild fasting hyperglycemia present at birth and may go undiagnosed until later in life [37]. Interestingly, heterozygous inactivating *GCK* variants cause permanent hyperinsulinemia/hypoglycemia conditions from birth [38], while patients with the homozygous form exhibit permanent neonatal diabetes [39]. Currently, there are over 600 mutations within the open reading frame of the *GCK* gene, and most of these variants are non-consistent within an ethnic group [40,41]. Most mutations are inherited, although de novo mutations have also been described [39,42,43]. 

GKAs are a class of small-molecule drugs that bind to an allosteric site on GK, stabilizing a high-affinity conformation and facilitating its activation [44]. Since 2001, pharmacological GKAs have been developed and assessed in animal models. However, several compounds have been discontinued due to AE profiles [45,46]. Early human clinical trials with GKAs demonstrated low efficacy and raised safety concerns, including hypoglycemia, liver steatosis, hypertriglyceridemia, systemic hypertension, and a failure to sustain long-term durability of the glucose-lowering effect [47,48], best reviewed in [49]. Therefore, pharmaceutical companies intensified efforts to design a new generation of GKAs. 

In the past few years, a novel generation of GKAs has been reported. Based on their mechanism of action, these GKAs have been classified into two major groups: dual-acting GKAs (such as dorzagliatin), which target both pancreatic and hepatic GK [50], and liver-selective GKAs (such as TTP399), which specifically target hepatic GK [51]. Furthermore, dual-acting GKAs are subclassified based on their kinetic effects on the GK as either “full” GKAs, increasing the enzymatic maximum velocity (Vmax) of GK, or as “partial” GKAs, reducing the Michaelis constant (Km) of GK [52]. Clinical trials have shown improved outcomes in terms of efficacy and the absence of AEs observed in earlier generations of GKAs. The current review aims to summarize the preclinical and clinical trials conducted on either dorzagliatin or TTP399. Only the trials with published results in PubMed are listed in chronological order of publication date through the end of 2023. We have also included, in a Appendix A, all the clinical trials listed on clinicaltrials.gov that involve either dorzagliatin, TTP399, or their alternative names. These trials were retrieved on 28 December 2023 (See Appendix A).

## 3. Dorzagliatin (Also Known as Sinogliatin, HMS-5552, or RO-5305552) 

### 3.1. Dorzagliatin: Mechanisms of Action

Dorzagliatin is a dual-acting GKA currently in the clinical developmental phase for treating T2DM. It is postulated that dorzagliatin binds to the allosteric site of GK at the P-loop, a conserved motif critical in regulating GK activity [53]. Th P-loop forms a pocket distal to the active site of GK, enhancing its affinity for glucose and lowering the set point for GSIS. This binding stabilizes a high-affinity conformation of the enzyme, increasing its activity and improving its ability to phosphorylate glucose. In the pancreas, dorzagliatin activates GK and exerts glucose-lowering effects by inhibiting IR and increasing insulin sensitivity (see Figure 1). Meanwhile, in the liver, dorzagliatin activates GK and promotes the dissociation of the GK-GKRP complex (see Figure 2) [1]. 

Dorzagliatin is a small-molecule drug with a molecular formula of C20H22N6O2, IUPAC name (2S)-2-(4-(2-Chloro-phenoxy)-2-oxo-2,5-dihydro-1H-pyrrol-1-yl)-N-(1-((2R)-2,3-dihydroxypropyl)-1H-pyrazol-3-yl)-4-methyl-pentanamide, and a molecular weight of 462.9 g/mol, appearing as a white-to-off-white solid with a melting point of 242–244 °C. Dorzagliatin is slightly soluble in water, but exhibits a higher level of solubility in organic solvents, ethanol, and DMSO. 

Structurally, dorzagliatin can be divided into three components. The central core comprises a pyrazole ring and a phenyl ring connected by an amide bond, critical for the drug’s binding affinity to GK. The substituent group is attached to the central core’s phenyl ring, consisting of an amino group and a guanidine group, mediating hydrogen bonding and electrostatic interactions with GK. The terminal alkyl chain is linked to the central core’s pyrazole ring, composed of an ethyl group and a methyl group, engaged in hydrophobic interactions with GK [53,54].

Herein, we will delve into preclinical studies and clinical trials employing dorzagliatin, summarizing the outcomes in Table 1.

### 3.2. Dorzagliatin: Preclinical Studies

Using male Sprague–Dawley rats, a preclinical study was conducted to evaluate the effectiveness of dorzagliatin in improving glucose metabolism. The rats were first induced to develop T2DM by administering a combination of a high-fat diet and streptozotocin. Subsequently, the diabetic rats were divided into four groups: a control group, a diabetic group, a group receiving 10 mg/kg of dorzagliatin, and a group receiving 30 mg/kg of dorzagliatin. After 27 days of treatment, dorzagliatin significantly reduced FPG levels by approximately 18% and 23% in the 10 mg/kg and 30 mg/kg treatment groups, respectively. Dorzagliatin treatment also modestly increased fasting glucagon (FG) levels while significantly decreasing fasting plasma insulin (FINS) levels. Oral glucose tolerance test (OGTT) results demonstrated a noticeable improvement in glucose tolerance in dorzagliatin-treated rats compared to untreated diabetic rats. Immunohistochemical analyses of dorzagliatin-treated rats showed a notably higher number of GK-immunopositive cells in the liver and insulin-immunopositive cells in the pancreas, as compared to that in untreated animal tissues. Western blot analysis further corroborated these findings, showing a significantly higher expression of GK proteins in the 30 mg/kg dorzagliatin-treated group than in the diabetic group. The results of GK gene expression aligned with those of the Western blot analysis. These findings suggest that dorzagliatin exerts beneficial effects on GK activity, IR, and glucose metabolism in diabetic rats. Further studies are warranted to validate these results in humans [1]. 

### 3.3. Dorzagliatin: Clinical Studies

In a phase I trial involving 60 healthy volunteers, dorzagliatin was found to be safe and well-tolerated at single doses ranging from 5 to 50 mg. The drug exhibited linear pharmacokinetics with minimal urinary metabolites, suggesting that renal excretion is a minor elimination pathway. Dorzagliatin demonstrated a dose-dependent reduction in FPG, reaching up to 31.49% at the maximum tested dose of 50 mg. Preprandial plasma samples showed no insulin secretion, while postprandial samples revealed a dose-dependent rise in insulin levels. Notably, a comparable incidence of AEs was observed in both the dorzagliatin and placebo groups, at 14.6% and 16.7%, respectively. The AEs were mild and included dizziness, palpitation, belching, cold sweats, and proteinuria [52].

In a randomized 4-week study involving Chinese T2DM patients with well-defined disease biomarkers, it was demonstrated that administering 75 mg dorzagliatin once (QD) or twice (BID) daily significantly reduced glycated hemoglobin (HbA1c) levels, FPG, 2-h post-prandial glucose (PPG), and 24-h glucose area under the curve (AUC). Dorzagliatin also substantially increased β-cell function, with HOMA2-B and ΔC30/ΔG30 increasing by approximately 40% and 168%, respectively. The drug was generally safe and well-tolerated, with mild hypoglycemia being the most common AE [55].

A 12-week phase II multicenter, randomized, double-blind, placebo-controlled study in China found that dorzagliatin is a safe and effective treatment for patients with T2DM. Patients who received dorzagliatin had dose-dependent reductions in HbA1c levels, with the 50 mg or 75 mg BID groups experiencing the most significant reductions. Almost half of the patients in these groups achieved the target HbA1c level of <7.0%. The 75 mg BID regimen also showed the highest decrease in FPG levels. Drug-naïve patients had a greater reduction in HbA1c levels than previously treated patients. The study recommended 75 mg BID as the minimum effective dose for achieving glycemic control with a favorable safety profile. After drug withdrawal, there were sustained improvements in the disposition index and HOMA for insulin resistance. Most treatment-emergent adverse events (TEAEs) were mild, including upper respiratory tract infections, hyperuricemia, and dizziness. There were similar rates of mild hypoglycemia across all dorzagliatin subgroups [50,56]

The phase III randomized, double-blind, placebo-controlled SEED clinical trial evaluated the long-term safety and efficacy of dorzagliatin in newly diagnosed, drug-naïve T2DM patients who failed to control their blood glucose levels with diet and exercise. Patients were randomly assigned to either the dorzagliatin or placebo group for 24 weeks, followed by 28 weeks of open-label dorzagliatin treatment for all patients. After 24 weeks of treatment, dorzagliatin significantly reduced HbA1c levels compared to those of the placebo group, and a higher proportion of dorzagliatin-treated patients achieved HbA1c levels of <7.0%. Dorzagliatin significantly reduced FPG and 2 h postprandial glucose (PPG) levels, while enhancing β-cell function. Throughout the open-label treatment period (weeks 24–52), the decline in HbA1c levels was consistently notable. Furthermore, the transition from placebo to dorzagliatin in the open-label period resulted in a significant reduction in HbA1c levels. The frequency of AEs was similar between the two groups, with no instances of severe hypoglycemia events, only one case, and no drug-related serious AEs observed in the dorzagliatin group. These observations suggest that dorzagliatin is a safe, well-tolerated, and effective long-term treatment for newly diagnosed T2DM patients [57,58].

The phase III randomized double-blind placebo-controlled DAWN clinical trial evaluated the efficacy of dorzagliatin as an add-on therapy in T2DM patients who had inadequate glycemic control with metformin monotherapy. After 24 weeks of treatment, dorzagliatin significantly reduced HbA1c levels compared to those of the placebo group. Approximately 44% of patients treated with dorzagliatin and metformin achieved an HbA1c level of <7%, compared to only 10% of patients treated with placebo and metformin. Dorzagliatin also significantly reduced FPG and PPG levels and improved β-cell function and insulin sensitivity. At the end of the open-label treatment period, weeks 24–52, reduced HbA1c levels were observed in the dorzagliatin and metformin group. Furthermore, switching from the placebo control to dorzagliatin and metformin dual therapy during the open-label period resulted in a significant reduction in HbA1c levels. Dorzagliatin was safe and well-tolerated, with a low incidence of hypoglycemia. These findings suggest that dorzagliatin as an add-on therapy is effective and safe [59]. 

A study involving patients with end-stage renal disease (ESRD) was conducted to evaluate the effects of renal impairment (RI) on the pharmacokinetics and safety of dorzagliatin. The study also investigated the appropriate dose for diabetes control in patients with diabetic kidney disease (DKD). Patients with ESRD were matched with healthy volunteers. The study concluded that systemic exposure to dorzagliatin is not clinically affected by ESRD, and no dose-adjustment was required for patients with DKD with various levels of RI. Dorzagliatin (25 mg QD) was well-tolerated in both the groups, i.e., non-dialysis ESRD patients and healthy volunteers. The TEAEs reported were mild, with only one case of mild hypoglycemia reported in a healthy volunteer. In the non-dialysis ESRD group, patients treated with dorzagliatin experienced headache and dry mouth and showed an increase in blood alkaline phosphatase [60].

A recent double-blind, randomized, crossover study investigated the effects of 75 mg dorzagliatin on insulin secretion rates (ISRs) and β-cell glucose sensitivity (βCGS) in participants with GCK-MODY and recent-onset T2DM. The study found that dorzagliatin lowered basal blood glucose levels in participants with GCK-MODY but had no effect on steady-state blood glucose levels in either group. In patients with T2DM, dorzagliatin increased ISRs compared to those of the placebo. Meanwhile, in patients with GCK-MODY, dorzagliatin increased βCGS compared to that of the placebo. In vitro studies on patients with GCK-MODY demonstrated that dorzagliatin restores glucose sensing in wild-type and mutant GK enzyme activity. Altogether, dorzagliatin was well-tolerated in both groups [61].

The DREAM study was a longitudinal follow-up of the SEED clinical trial that was conducted to investigate the long-term effects of dorzagliatin in newly diagnosed patients with T2DM who had previously achieved stable glycemic control and sustained disease remission. In this study, T2DM remission was achieved with a notable improvement in β-cell function, insulin sensitivity, and time in range (TIR). Additionally, the Kaplan–Meier and the American Diabetes Association (ADA) remission probabilities were 65.2% (at week 52) and 52.0% (at week 12), respectively. Dorzagliatin emerged as an effective and safe treatment for newly diagnosed T2DM patients. In conclusion, dorzagliatin treatment demonstrated the potential to achieve sustained glycemic control and drug-free diabetes remission in these patients [62].

A recent phase I clinical trial was conducted to evaluate the safety, pharmacokinetics, and pharmacodynamics of dorzagliatin when co-administered with sitagliptin in patients with obesity and T2DM. The trial found that there were no clinically meaningful pharmacokinetic interactions between dorzagliatin and sitagliptin. The combination treatment was well-tolerated and did not increase the incidence of AEs. Additionally, the combination treatment demonstrated improved glycemic control compared to that of either monotherapy. These findings suggest that the co-administration of dorzagliatin and sitagliptin may represent a promising treatment option for patients with T2DM and obesity [63].

## 4. TTP399 (Also Known as Cadisegliatin and GK1-399)

### 4.1. TTP399: Mechanisms of Action

TTP399 (Vtv Therapeutics LLC, Wixom, MI, USA) is an orally active, liver-selective GKA [64]. It has demonstrated improved glycemic control in both animal models and patients with T2DM by activating the hepatic GK [51]. The structural interaction between GK and TTP399 has not been definitively determined through direct structural modeling. However, a structurally similar GKA was co-crystallized with GK, providing insights into the potential binding mode of TTP399. The molecule is called 17c, it expands the glucokinase binding cavity, which is embedded within the β1 strand and R5 helix of the glucokinase large domain, the C-terminal R13 helix of the small domain, and the glucokinase-specific connecting region I [65]. Therefore, it is postulated that TTP399 binds to the allosteric site of GK, inducing a conformational change in the protein that enhances its catalytic activity, without disrupting the interaction between GK and GKRP (Figure 3) [51].

TTP399 is a small-molecule drug with a molecular formula of C_21_H_33_N_3_O_4_S_2_, the IUPAC name 2-[[2-[[cyclohexyl-(4-propoxy-cyclohexyl)-carbamoyl]-amino]-1,3-thiazol-5-yl]-sulfanyl]-acetic acid, and a molecular weight of 455.6 g/mol. It is an off-white to pale purple solid with a melting point of 206–207 °C. It is insoluble in water and has low solubility in most organic solvents. 

In the following parts, we will examine both preclinical studies and clinical trials utilizing TTP399, with a summary of the results presented in Table 2.

### 4.2. TTP399: Preclinical Studies

TTP399 was evaluated in several preclinical studies to assess its impact on glucose metabolism and insulin secretion. In vivo studies using Wistar rats, mice, and Gottingen minipigs demonstrated that TTP399 does not activate glucokinase in pancreatic β-cells, does not alter insulin secretion, and does not result in hypoglycemia [51]. Instead, it effectively increases glucose metabolism by enhancing lactate and glycogen production. Animal studies on rat hepatocytes suggest that TTP399 preserves the physiological regulation of glucokinase through GKRP, ensuring that the liver maintains control over glucose metabolism [51]. This preservation of the GK–GKRP interaction appears to be unique to TTP399. The preclinical data indicate that TTP399 improves glycemic control, reduces IR, and decreases body weight without affecting plasma insulin levels or hepatic lipids. In animal models with dyslipidemia and a fatty liver, TTP399 administration also reduced triglyceride concentrations in both the plasma and liver [51].

### 4.3. TTP399: Clinical Studies

The Simplici-T1 clinical trial evaluated the safety and efficacy of TTP399 as an adjunctive therapy for type 1 diabetes mellitus (T1DM) [66,67]. A total of 109 participants were enrolled, who were randomized to receive either TTP399 or placebo QD for 12 weeks. Part 1 of the trial involved 19 patients on insulin pump therapy and continuous glucose monitoring, whereas part 2 of the trial involved 85 patients with a broader range of T1DM severity. In both the study arms, TTP399 significantly reduced HbA1c levels and bolus insulin compared to those in the placebo group [66]. Additionally, TTP399 improved daytime TIR (ITT: 8%, 95% CI 1.15, *p* < 0.01, HC) and had a lower incidence of TEAE compared to those of the placebo group. In both parts 1 and 2, plasma β-hydroxybutyrate and urinary ketone levels were lower during treatment with TTP399 than with the placebo. Moreover, no AEs were identified [66]. Overall, the Simplici-T1 trial demonstrated that TTP399 is a safe and effective adjunctive therapy for T1DM, and further development of this drug is warranted [66].

The Glucokinase Activator to Target HbA1c (AGATA) clinical trial evaluated the efficacy of TTP399 over a 6-month period in 190 patients with T2DM who were receiving a stable dose of metformin [51]. The primary outcome of the study was a placebo-subtracted change in HbA1c from the baseline, which was −0.9% with TTP399 (800 mg), −0.2% with TTP399 (400 mg), and −1.0% with sitagliptin (100 mg). TTP399 was not associated with changes in the overall weight. However, in patients weighing ≥ 100 kg at baseline, a weight loss of 3.4 kg was noted with TTP399 (800 mg/day). Patients receiving TTP399 (800 mg/day) also experienced a significant decrease in FG concentrations (−19.6 pg/mL) relative to the baseline placebo. Additionally, there was a remarkable increase in HDL-C (+3.2 mg/dL) in patients treated with TTP399 (800 mg/day) versus the placebo group. The percentage of individuals who developed TEAEs was generally similar among groups (approximately 56%) [51]. No serious AEs were reported, and AEs leading to study withdrawal were rare. The most frequently reported AEs that occurred in ≥ 5% of patients were headache, upper respiratory tract infection (URTI), diarrhea, nausea, hypoglycemia, urinary tract infection, cough, and nasopharyngitis. No cases of severe hypoglycemia were reported [51]. Overall, the AGATA clinical trial demonstrated that TTP399 is a safe and effective treatment for T2DM. The drug significantly reduced HbA1c levels and improved HDL-C levels without causing weight gain or hypoglycemia. Further studies are needed to confirm these findings in larger and longer-term trials.

A clinical study investigated the effect of TTP399 on the risk of ketoacidosis during insulin withdrawal in T1DM patients [68]. Participants with T1DM using insulin pump therapy were randomized to receive either 800 mg TTP399 or a placebo for 7 to 10 days. Following treatment, an insulin withdrawal test (IWT) was conducted to induce ketogenesis. The primary endpoint was the proportion of patients who reached beta-hydroxybutyrate (BHB) concentrations of 1 mmol/L or greater. During the treatment period, TTP399 significantly reduced mean FPG levels and resulted in fewer AEs compared to placebo treatment [68]. During the IWT, there were no differences between the TTP399 and placebo groups in the mean serum BHB concentration, mean duration of IWT, or BHB at termination of IWT. However, serum bicarbonate was statistically higher and urine acetoacetate was quantitatively lower in the TTP399-treated patients [68]. Importantly, none of the TTP399-treated participants met the criteria for DKA, as compared to 42% of the placebo-treated individuals. In conclusion, TTP399 effectively improves glycemia without increasing hypoglycemia in individuals with T1DM. Additionally, TTP399 does not increase BHB concentrations and reduces the incidence of DKA during acute insulin withdrawal. These findings suggest that TTP399 may be a promising adjunctive therapy for T1DM [68].

## 5. Discussion

The increasing prevalence of T2DM and the inadequacy of current oral hyperglycemic drugs to achieve sufficient glycemic control have prompted the development of novel therapeutics. Due to its progressive nature, monotherapy with metformin has limited tolerability and efficacy for the treatment of diabetes, ultimately necessitating combination therapy [13]. Furthermore, the currently available and prescribed oral hypoglycemic drugs are linked to severe AEs and an eventual loss of efficacy. 

GK activation presents an alternative approach to improve glycemic control in patients with T2DM. The older-generation GKAs faced several challenges, including hypoglycemia, dyslipidemia, a lack of long-term efficacy, and toxicity. However, the development of the new-generation, dual-acting dorzagliatin and hepato-selective TTP399 stimulates insulin and glucagon secretion and promotes glycogen synthesis and storage, addressing the shortcomings in terms of efficacy and safety.

A meta-analysis and trial sequential analysis (TSA) incorporating three clinical studies—phase II, phase III (SEED and DAWN trials)—confirmed the benefits of dorzagliatin in lowering HbA1c, FPG, and 2h-PPG. Additionally, TSA affirmed an increase in HOMA2-b and a decrease in HOMA2-IR [69]. Results from these trials suggest that dorzagliatin can be administered to patients with mild-to-moderate diabetes. However, regarding blood lipid-related endpoints, the meta-analysis revealed that dorzagliatin increased triglycerides by 0.43 mmol/L and total cholesterol by 0.13 mmol/L, as compared with the placebo levels, which was further validated by TSA. This necessitates additional investigations and trials due to the association of increased triglycerides with T2DM, posing an elevated cardiovascular risk. In terms of safety, despite an overall higher incidence of total AEs in individuals treated with the dual agonist GKA, both the meta-analysis and TSA confirmed that dorzagliatin does not elevate serious AEs. The observed adverse reactions might be attributed to a combination of single AEs and intrinsic individual characteristics, thereby justifying its favorable safety profile [69]. A notable advantage of dorzagliatin is its safety in controlling hyperglycemia in patients with DKD or renal insufficiency. Achieving glycemic control in patients with DKD, one of the complications of diabetes, is particularly challenging, as commonly used treatments are either contraindicated, not recommended, or necessitate dose adjustment with frequent monitoring of renal function [60].

One of the most commonly inherited diabetes subtypes, GCK-MODY, exhibits a notable reduction in βCGS and impaired α-cell glucose sensing. Currently, there are no effective glucose-lowering drugs to address the defects of MODY. Dorzagliatin demonstrated improvement in ISR and β-cell function in patients with selected GK mutations through recent in vivo and in vitro studies [61]. Despite challenges such as a limited sample size, different mutations, and a varied intrinsic β-cell secretory capacity, this study holds potential value in precision medicine [61].

In the context of treatment with TTP399, a reduction in HbA1c of 0.9% and in FG of 20 pg/mL from baseline was observed. Additionally, TTP399 significantly increased HDL-C levels and induced weight loss (3.4 kg) in patients over 100 kg [64]. While weight loss was observed in a specific cohort of individuals, TTP399 provides a weight-neutral effect, unlike some current drugs that tend to cause weight gain (e.g., sulfonylureas and thiazolidinediones) [13]. Throughout the AGATA trial, treatment with TTP399 was found to be well-tolerated, effective, and safe, with mild TEAEs, such as URTI, headache, diarrhea, and nausea [64]. 

While both dorzagliatin and TTP399 have shown outstanding results in terms of their efficacy and safety in treating T2DM, dorzagliatin seems to have the upper hand. Dorzagliatin has proved to be beneficial in a wide spectrum of patients with diabetes, along with those affected by renal insufficiency due to DKD. Moreover, dorzagliatin successfully completed two phase III clinical trials that were conducted for a total of 52 weeks, as compared to the phase II trial of TTP399, which lasted approximately 25 weeks. Additional research is required to establish its role as a monotherapy or combination treatment. Furthermore, studies involving dorzagliatin should encompass diverse ethnicities, as most of the clinical trials were conducted on people of Chinese descent. Conversely, a phase II study, called the Simplici-T1 study, evaluated the safety and efficacy of TTP399 in subjects with T1DM, as it could be a breakthrough oral hypoglycemic therapy for T1DM.

Notably, while the potentials of dorzagliatin and TTP399 in diabetes are promising, conducting further clinical studies is crucial to fully evaluate their efficacy and safety when compared to established combinational therapies. Investigating the effectiveness of these new GKAs in patients with common comorbidities such as cardiovascular disease, heart failure, and renal impairment is particularly important, given their high prevalence and significant contribution to morbidity and mortality, often driven by impaired metabolic homeostasis. Considering the well-established role of GKAs in regulating glucose metabolism and their safety in trials involving patients with ESRD and cardiovascular disease, we anticipate that co-administering dorzagliatin in particular or TTP399 with medications for metabolic diseases in these patients may hold the promise of a synergistic therapeutic effect, potentially leading to improved treatment outcomes.

## 6. Conclusions and Future Perspectives

Dorzagliatin and TTP399 are two GKAs that hold promise as potential treatments for diabetes mellitus. Further research is needed to elucidate detailed mechanisms of action and identify ways to enhance their efficacy or develop even more effective novel therapies. Additionally, there is a growing need for diabetes mellitus interventional trials focused on restoring functional β-cell mass to improve disease management and prevention. These trials should consider incorporating immunomodulatory interventions and prospective precision medicine approaches. Large, multicenter, multi-ethnic studies involving electronic health data monitoring could help determine whether the GKAs effectively prevent diabetes progression. Clinical trials should also encompass participants with newly diagnosed disease and those with a long history of diabetes, individuals with prediabetes, and those with metabolic syndromes. Researchers working on GKA therapies are also encouraged to conduct long-term longitudinal follow-up studies. The involvement of international societies and organizations would be highly beneficial in establishing guidelines and consensus statements for the safe practice of GKA therapies in diabetes mellitus.

## Figures and Tables

**Figure 1 ijms-25-00571-f001:**
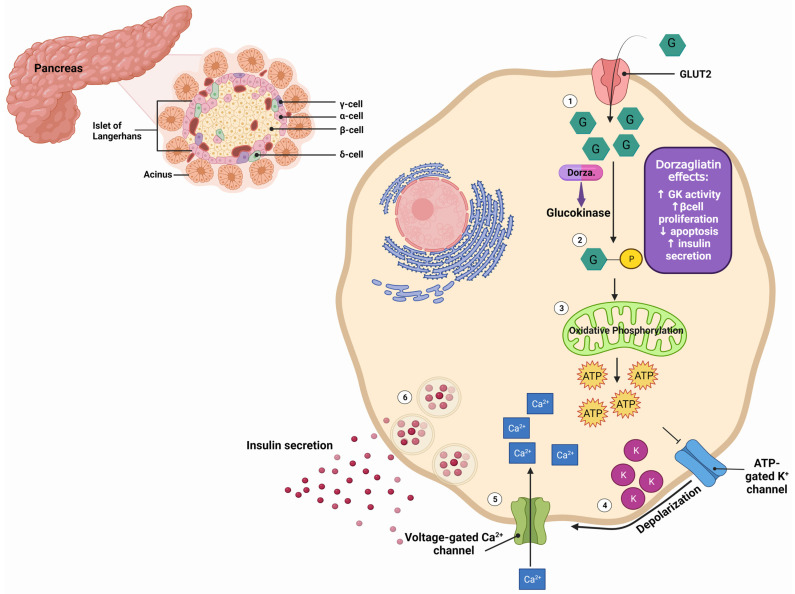
Effects of dorzagliatin on the pancreas: As blood glucose levels increase (1), dorzagliatin directly activates glucokinase (2), a crucial enzyme in the glycolysis pathway. Subsequent oxidative phosphorylation (3) generates ATP. The elevated ATP levels cause the closure of ATP-sensitive potassium channels (4), leading to membrane depolarization. This depolarization triggers the opening of voltage-gated calcium channels (5). The increased intracellular calcium concentration stimulates the exocytosis of insulin-containing secretory vesicles from the β-cells, leading to the release of insulin from the pancreatic β-cells (6). Moreover, dorzagliatin stimulates the proliferation and limits apoptosis of β-cells, thereby improving blood glucose levels. GK, glucokinase; GLUT2, glucose transporter 2; F cells, pancreatic polypeptide cells; α-cell, pancreatic alpha cells; β-cell, pancreatic beta cells; δ-cell, pancreatic delta cells. Created with BioRender.com.

**Figure 2 ijms-25-00571-f002:**
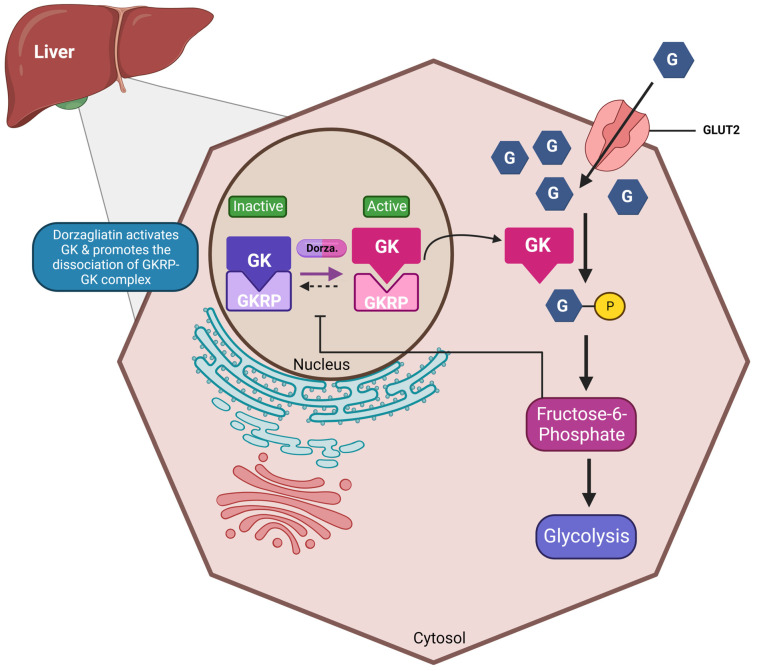
Effects of dorzagliatin on the liver: Under normoglycemic conditions, glucokinase is tightly bound to GKRP in the nucleus. As blood sugar levels rise, dorzagliatin promotes the dissociation of the GK-GKRP complex through the direct activation of GK and translocates it to the cytoplasm, initiating the glycolysis pathway. The energy generated can then be further utilized for glycogenolysis. GKA, glucokinase activator; GK, glucokinase; GKRP: glucokinase regulatory protein; GLUT2, glucose-transporter 2. Created with BioRender.com.

**Figure 3 ijms-25-00571-f003:**
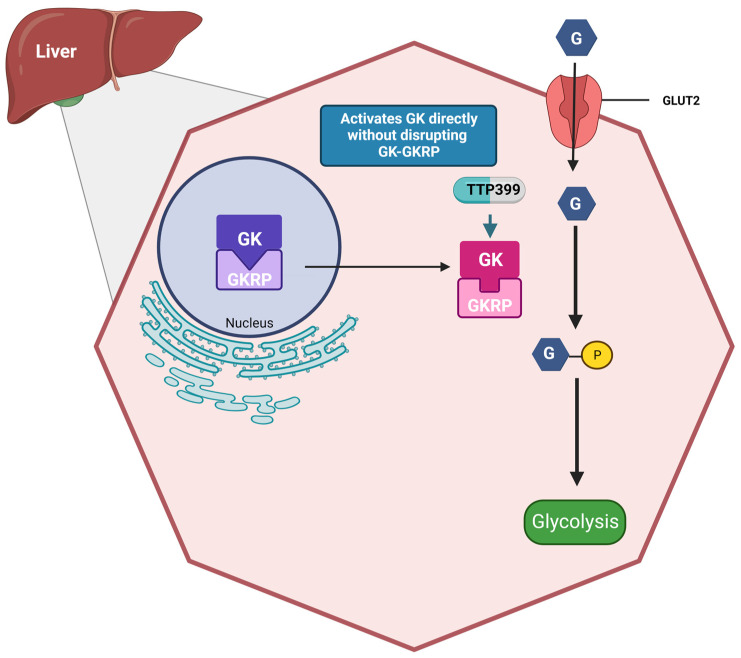
Effects of TTP399 on the liver: TTP399 is a liver-specific glucokinase activator that directly activates glucokinase in the cytosol without disrupting the GK-GKRP complex. This hepatocyte-specific drug mitigates the risk of hypoglycemia that was associated with previously manufactured glucokinase activators. G, glucose; GK, glucokinase; GKRP, glucokinase regulatory protein; GLUT2, glucose transporter 2. Created with BioRender.com.

**Table 1 ijms-25-00571-t001:** Results from preclinical studies and clinical trials using dorzagliatin.

Preclinical Studies					
Reference	Animal	Duration (Weeks)	Interventions	Primary Findings	
2017, Wang et al. [1]	Male Sprague–Dawley (SD) rats with T2DM	4	Control, diabetic, 10 mg/kg, 30 mg/kg	Reduction in FPG by ∼18% (10 mg/kg) and 23% (30 mg/kg) Reduction in FINS: 28.40 mU/L (10 mg/kg) and 18.74 mU/L (30 mg/kg) Levels of TC and TG unchanged Increase in FG: 43 pg/mL (10 mg/kg) and 51 pg/mL (30 mg/kg) Reduction in OGTT: 9 mmol/L (10 mg/kg) and 7 mmol/L (30 mg/kg) compared to diabetic rats Increased expression of GK-immunopositive cells and insulin (Western blot)	
**Clinical Trials**					
**Reference**	**Total Participants (N)**	**Duration (weeks)**	**Study design**	**Interventions**	**Primary findings**
2016, Xu et al. [52]	60 (31 M, 29 F)	XXX	Phase Ia: randomized, double-blind, placebo-controlled, parallel-group, administered to healthy subjects	Six dose cohorts (5, 10, 15, 25, 35, and 50 mg), 10 randomized subjects (8 receiving HMS5552 and 2 receiving placebo)	HMS5552 at doses up to 50 mg in healthy subjects is safe and well-tolerated Dose-related glucose-lowering effects and post-prandial insulin secretion AEs: belching, dizziness, palpitation, cold sweat, and proteinuria No hypoglycemia
2018, Zhu et al. [55]	24 (17 M, 7 F)	4	T2DM patients randomized at 1:1 ratio to receive two concentrations of dorzagliatin	75 mg QD, 75 mg BID	Overall, QD treatment was better than BID Decrease in HBA1c, FPG and PPG Increase in C-peptide Reduction in AUC Increase in HOMA2 parameter %B Increase in the dynamic state parameter ΔC30/ΔG30 Hypoglycemia: 17%
2018, Zhu et al. [50,56]	258 (154 M, 104 F)	12	Phase II: multicenter, randomized, double-blind, placebo-controlled T2DM patients were on a diet and exercise regimen; drug-naïve or previously treated with metformin or α-glucosidase inhibitor monotherapy	75 or 100 mg QD, 50 or 75 mg BID, placebo	Decrease in HbA1c, FPG, and PPG, particularly in patients who received 100 QD or 75 BID AE: URTi, hyperuricemia, dizziness Hypoglycemia 6% in all studied groups
2020–2022, Zhu et al. (SEED trial) [57,58]	463 (301 M, 162 F)	52	Phase III: multicenter, randomized, double-blind, placebo-controlled (24 weeks), open-label (28 weeks) study T2DM drug-naïve patients	75 mg BID, placebo	Decrease in HbA1c and FPG at week 24, sustained through week 52 Increase in HOMA2-β AEs: URTi, hyperlipidemia, proteinuria, abnormal hepatic function, hypertension Hypoglycemia 0.3% in all studied groups
2022, Yang et al. (DAWN trial) [59]	767 (475 M, 292 F)	52	Phase III: randomized, double-blind, placebo-controlled (24 weeks), open-label (28 weeks) study T2DM add-on therapy to metformin	75 mg BID, placebo, add-on metformin 1500 mg	Decrease in HbA1c < 7% in 44.4% patients at week 24 Decrease in FPG and HOMA2-IR Increase in HOMA2-β Hypoglycemia 0.3%, no weight gain in all studied groups
2022, Miao et al. [60]	17 (7 M, 10 F)	½	Open-label, single-dose, sequential two-part, parallel-group study 8 non-dialysis ESRD, including 1 T2DM; 9 healthy volunteers	25 mg QD	End-stage renal insufficiency does not affect dorzagliatin efficacy Dorzagliatin absorption is rapid peak plasma concentration (Cmax) 1.25–2.5 h post-dose Dorzagliatin elimination half-life (t1/2) for dorzagliatin is 4.5–8.6 h
2023, Chow et al. [61]	18 (6 M, 12 F)	2	Phase II: randomized, double-blind, cross-over study 8 GCK-MODY; 10 T2DM	Single dose First group: 75 mg dorzagliatin (first visit), placebo (second visit) Second group: Placebo (first visit), 75 mg dorzagliatin (second visit)	GCK-MODY, dorzagliatin significantly increased absolute and incremental second-phase ISRs Dorzagliatin improves β-cell glucose sensitivity in GCK-MODY Dorzagliatin increases basal prehepatic insulin secretion rates in T2DM Dorzagliatin restores GK enzymatic activity
2023, Zeng et al. [62] (DREAM, longitudinal SEED study)	69 (48 M, 21 F)	52	Phase III: randomized, double-blind, placebo-controlled (24 weeks), open-label (28 weeks) study Patients who completed the SEED trial (56), who achieved stable glycemic control with potential to sustain drug-free remission Glycemic control status was assessed at weeks 0, 12, 26, 39, and 52.	T2DM remission, glycemic control status was assessed at weeks 0, 12, 26, 39, and 52.	Dorzagliatin leads to stable glycemic control and drug-free remission in drug-naïve patients with T2DM Dorzagliatin sustains HbA1c and FPG levels and glucose homeostasis Dorzagliatin maintains the steady-state β-cell function and insulin resistance
2023, Chen et al. [63]	15	2	Phase I: open-label, single-sequence, multiple-dose, single-center Dorzagliatin + sitagliptin in obese T2DM patients	Day 1–5: sitagliptin 100 mg QD on Day 1–5 Day 6–10: sitagliptin 100 mg QD and dorzagliatin 75 mg BID Day 11–15: dorzagliatin 75 mg BID alone	Combination treatment did not increase AEs and was well-tolerated in T2DM No pharmacokinetic interactions between dorzagliatin and sitagliptin Improvement of glycemic control under combination conditions

ΔC30/ΔG30, dynamic state parameter; %B, steady-state HOMA 2 parameter; AE, adverse event; AUC, last quantifiable concentration; BID, twice a day; Cmax, maximum plasma concentration; F, female; FG, fasting glucagon; FPG, fasting plasma glucose; GK, glucokinase; HOMA2, Homeostatic Model Assessment 2; IR, insulin resistance; M, male; OGTT, oral glucose tolerance test; PPG, post-prandial glucose; T2DM, type 2 diabetes mellitus; TC, total cholesterol; TG, triglyceride; URTI, upper respiratory tract infection; QD, once a day.

**Table 2 ijms-25-00571-t002:** Results from preclinical studies and clinical trials using TTP399.

Preclinical Studies					
Reference	Animal	Duration (Weeks)	Interventions	Primary Findings	
2019, Vella et al. (AGATA trial) [51]	Umea ob/ob mice with T2DM	4	75, 150 mg/kg	Reduction in HbA1c from baseline (week 4) 0.76 ± 0.14% (75 mg/kg); 1.23 ± 0.28% (150 mg/kg) Lower blood glucose Reduced weight gain 0.5 g with 150 mg/kg per day Reduced TG concentrations	
2019, Vella et al. (AGATA trial) [51]	Gottingen minipigs with T2DM	13	50 mg/kg ± add-on metformin 500 mg BID	Reduced plasma glucose levels	
**Clinical Trials**					
**Reference**	**Total Participants (N)**	**Duration (Weeks)**	**Study design**	**Interventions**	**Primary findings**
2019, Vella et al. (AGATA trial) [51]	190 (101 M, 89 F) T2DM	24	Phase II b, randomized, double-blind, placebo and active-controlled, parallel group	TTP399 400 mg QD TTP399 800 mg QD Sitagliptin 100 mg QD Placebo ± add-on metformin	Reduction in HbA1c from baseline Decrease in FG concentrations Increase in HDL-C Lower blood glucose No change in body weight, no hypoglycemia
2021, Klein K.R. et al. (The Simplici-T1 trial) [66,67]	Part 1: 20 (7 M, 13 F) Part 2: 85 (47 M, 38 F) All T2DM	12	Phase 1b/2, randomized, double-blind, placebo-controlled Part 1: T1DM on insulin pump therapy and continuous glucose monitoring (CSII) Part 2: T1DM with a broader range of T1DM severity, multiple daily injections of insulin or CSII.	TTP399 800 mg QD Placebo	Reduction in HbA1c, improved daytime TIR Reduced plasma β-hydroxybutyrate and urinary ketones No adverse events
2022, Klein K.R. et al. [68]	23 (10 M, 13 F) T1DM	1–1½	Phase 1, double-blind, randomized, parallel-grouped, placebo-controlled multiple-dose study T1DM using insulin pump therapy	IWT test, ketogenesis induction, TTP399 800 mg QD	Reduction in FPG Elevated serum bicarbonate, lower urine acetoacetate No increase in BHB concentrations Reduction in the incidence of DKA during acute insulin withdrawal.

AGATA, Add Glucokinase Activator to Target A1c; QD, once a day; BID, twice a day; CSII, continuous glucose monitoring; FPG, fasting plasma glucose; FINS, fasting insulin; FG, fasting glucagon; HBA1c, glycated hemoglobin; T2DM, type 2 diabetes mellitus; T1DM, type 1 diabetes mellitus; TC, total cholesterol; TG, triglycerides; TIR: time in range.

## Data Availability

Not applicable.

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
