# Peer review of "New-Generation Glucokinase Activators: Potential Game-Changers in Type 2 Diabetes Treatment"

_ijms, 2024, doi:10.3390/ijms25010571_

Round 1
Reviewer 1 Report
Comments and Suggestions for Authors
This manuscript presents a review about the mechanisms of action of new-generation glucokinase activators (GKAs) assessing their efficacy and safety in treating type 2 diabetes mellitus (t2 DM) based on published preclinical studies and clinical trials.
The manuscript seems well written, the language is correct throughout the text, and the references are in general up-to-date.
However, i would like that the authors add in the end of the discussion a paragraph of 2-3 sentences-lines on their own opinion of dorzagliatin role and if possible its efficacy on chronic kidney disease, and other comorbidities, such as cardiovascular disease and heart failure, which represent a significant health burden in the US and worldwide, and a main cause of morbidity and mortality.
Author Response
We appreciate the reviewer’s complements, and thanks for the valuable suggestions. A paragraph has been added in lines 438 to 448.

Reviewer 2 Report
Comments and Suggestions for Authors
In the submitted review the authors revise the literature related to new generation glucokinase activators (in particular Dorzagliatin and TTP399) highlighting their structure, the potential mechanism of action, the results of preclinical and clinical studies, outlining the advantages and limitations.
The topic is interesting, up to date and relevant in the context of basic and applied research for the treatment of diabetes. The work is appropriate for IJMS. The review is well organized and balanced. The tables are necessary and clear. I just have a quick comment on figures. Now it is not easy to grasp the mechanism of action of drugs and the added value of the treatments. For example, in fig 1 you could use a symbol (purple colour) to indicate the drug (currently there is only an arrow) and report "Effects of Dorzagliatin" in the purple box.
Please also include a short paragraph on how the papers and preclinical/clinical trials cited in the review were identified and selected.
Author Response
We would like to thank the reviewer for the suggestion. Figures 1 and 2 have been modified accordingly. We have included a short paragraph in lines 96 to 101. Only the trials with published results in PubMed are listed in chronological order of publication date through the end of 2023. Additionally, all clinical trials listed on clinicaltrials.gov involving either Dorzagliatin or TTP399 or their alternative names have been included in a supplementary file. These trials were retrieved on Dec 28, 2023.
